# Safety and Immunogenicity of a Carbohydrate Fatty Acid Monosulphate Ester Adjuvant Combined with a Low-Dose Quadrivalent Split-Virion Inactivated Influenza Vaccine: A Randomised, Observer-Blind, Active-Controlled, First-in-Human, Phase 1 Study

**DOI:** 10.3390/vaccines12091036

**Published:** 2024-09-10

**Authors:** Valentino D’Onofrio, Sharon Porrez, Bart Jacobs, Azhar Alhatemi, Fien De Boever, Gwenn Waerlop, Els Michels, Francesca Vanni, Alessandro Manenti, Geert Leroux-Roels, Peter Paul Platenburg, Luuk Hilgers, Isabel Leroux-Roels

**Affiliations:** 1Center for Vaccinology (CEVAC), Ghent University and Ghent University Hospital, Corneel Heymanslaan 10, 9000 Ghent, Belgium; valentino.donofrio@uzgent.be (V.D.); sharon.porrez@ugent.be (S.P.); bart.jacobs2@uzgent.be (B.J.); azhar.alhatemi@uzgent.be (A.A.); fien.deboever@uzgent.be (F.D.B.); gwenn.waerlop@ugent.be (G.W.); geert.lerouxroels@ugent.be (G.L.-R.); 2Harmony Clinical Research BV, 9090 Melle, Belgium; e.michels@harmony-cr.com; 3VisMederi S.r.l., 53035 Monteriggioni, Italy; francesca.vanni@vismederi.com (F.V.); alessandro.manenti@vismederi.com (A.M.); 4LiteVax, 4061 BJ Ophemert, The Netherlands; peterpaul.platenburg@litevax.com (P.P.P.); luuk.hilgers@litevax.com (L.H.)

**Keywords:** adjuvant, influenza vaccine, dose-sparing, safety, immunogenicity

## Abstract

Seasonal influenza vaccine effectiveness is low. Carbohydrate fatty acid monosulphate ester (CMS), a new oil-in-water adjuvant, has proven potency in animal models with suggested capacity for dose-sparing. The objective was to evaluate safety and immunogenicity of CMS when added to a low-dose influenza vaccine (QIV) in humans. In a randomised, double-blind, active-controlled, first-in-human study, sixty participants (18–50 years) received either 0.5 mg CMS or 2 mg CMS with 1/5th dose QIV, or a full dose QIV without CMS. Adverse events (AE) were monitored until 7 days post-vaccination. Haemagglutinin inhibition (HI) titres in serum and CD4+ T cells in PBMCs were determined at day 0, 7, 28, and 180. Mean age was 37.6 (±10.1) years and 42/60 (70.0%) were female. Pain at injection site (42/60, 86.7%) and headache (34/60, 56.7%) were reported most and more frequently in the 2 mg CMS group. HI titres and the frequency of influenza specific CD4+ T cells were equal across strains for the three cohorts on all visits, increased until day 28 and decreased at day 180 to values higher than baseline. CMS was safe in humans. Humoral and cell-mediated immunogenicity was similar across vaccines, even with 1/5th antigen dose. CMS can have beneficial implications in low-resource settings or in a pandemic context.

## 1. Introduction

Seasonal influenza is a contagious respiratory infection caused by either influenza A or B virus, capable of inducing mild or severe illnesses, including hospitalisation and, in some cases, fatalities. Some people, such as older adults, children under 5 years old, individuals with chronic illnesses, and pregnant women, are at higher risk of developing severe illness and complications. Vaccination remains the most effective method for preventing serious flu-related outcomes, including death [1].

Conventional influenza vaccines typically contain inactivated viruses, haemagglutinin (HA)/neuraminidase (NA) subunits or live attenuated viruses. One widely used vaccine is VaxigripTetra^®^ (Sanofi Pasteur, Lyon, France), a quadrivalent inactivated split virion influenza virus vaccine. The 2022–2023 seasonal vaccine contains 15 µg HA of each of the two influenza type A virus subtypes (A/Darwin (H3N2) and A/Victoria (H1N1)) and the two influenza B virus subtypes (B/Austria and B/Phuket), compliant with the WHO recommendations for the 2022/2023 season. This vaccine is known to induce superior immunogenicity compared to trivalent inactivated influenza vaccines as assessed by HI titres [2].

Despite the widespread availability and administration of influenza vaccines over the years, influenza continues to pose a significant public health threat globally, with millions of severe cases and hundreds of thousands of deaths reported annually [1,3,4,5]. Annual vaccination campaigns are crucial and are considered the most cost-effective strategy to control influenza.

However, licensed influenza vaccines often exhibit overall low (10–60%) vaccine effectiveness (VE), especially among high-risk populations such as older adults [5,6]. Alternative approaches, including live attenuated vaccines, higher antigen doses, and the use of adjuvants, have been explored to enhance vaccine efficacy [6,7,8]. Hence, there is still a pressing need for more efficacious influenza vaccines to mitigate illness severity and prevent serious outcomes like hospitalisation and death. Improved adjuvants hold promise to enhance VE.

Moreover, adopting single-low-dose approaches can enhance vaccine availability, accessibility, and affordability globally. By substituting a significant portion (80%) of costly antigen with an inexpensive adjuvant, vaccine production costs can be substantially reduced, making the product more accessible in low- and middle-income countries [7,9].

To address these challenges and boost VE, a novel experimental adjuvant has been developed. The adjuvant, belonging to a new class of synthetic carbohydrate fatty acid monosulphate esters (CMS), is immobilised on the surface of oil droplets within a nano-emulsion of squalane-in-water. Several studies in non-rodent species have demonstrated the potent efficacy of single-shot vaccines containing the CMS-adjuvant, showing comparable or superior antibody titres compared to two immunisations with MF59-adjuvanted vaccines [10]. In ferrets, CMS strongly enhanced humoral responses to a low dose of a licensed vaccine following both initial and booster immunisations [11]. Consequently, CMS may provide benefits like dose-sparing and improved immunogenicity when administered to older adults.

In this study, CMS was combined with a low (1/5th) dose of a licensed quadrivalent seasonal inactivated split virion influenza virus vaccine (VaxigripTetra). The aim was to assess the safety of CMS and evaluate its potency in stimulating influenza-specific immune responses when administered alongside a reduced dose of the influenza vaccine.

## 2. Methods

### 2.1. Study Design and Participants

This study is a randomised, observer-blind, active-controlled phase 1 trial conducted at a single centre, the Center for Vaccinology (CEVAC, Ghent University Hospital and Ghent University in Ghent, Belgium) between 26 September 2022 and 17 May 2023. Participants in the study were healthy male and female adults aged 18 and 50 years who had not had a confirmed influenza virus infection in the 12 months preceding study vaccination and had not received an influenza vaccination in the preceding 6 months. To further minimise the likelihood of unusually high baseline HI titres, vaccinations were administered in October–November 2022, which is the recommended period for influenza vaccinations in Belgium, just before the start of the seasonal epidemic and when virus circulation is low. Detailed inclusion and exclusion criteria can be found in Appendix A. All study procedures adhered to ICH and GCP guidelines. The study documents were approved by an independent Ethics Committee and the Belgian Federal Agency for Medicines and Health Products (FAMHP) (EudraCT number: 2022-500681-98-00). Written informed consent was obtained from all participants.

### 2.2. Study Vaccine

The adjuvant formulation comprises CMS combined with a squalane-in-water emulsion. It is a fully synthetic, sterile, aqueous, ready-for-use product characterised by its physical and chemical stability. During the course of this study, CMS was added to VaxigripTetra, a licensed quadrivalent seasonal inactivated split influenza virus vaccine, immediately prior to intramuscular administration. A reduced dose, equivalent to one-fifth (3 µg HA/strain) of the vaccine, was mixed with either 0.5 mg or 2 mg CMS and compared against the standard dose (15 µg of HA/strain) of VaxigripTetra. The VaxigripTetra vaccine used was the 2022–2023 formulation, composed according to WHO’s recommendations of that season [7]. All three vaccine formulations were administered intramuscularly as a 0.5 mL dose.

### 2.3. Study Procedures

Sixty healthy participants were randomly assigned in an observer-blind manner to three cohorts. In each cohort, 20 individuals received one of the following vaccines in the deltoid muscle of the non-dominant arm: full dose VaxigripTetra (15 µg) (control vaccine), low dose VaxigripTetra (3 µg) + 0.5 mg CMS, or low dose VaxigripTetra (3 µg) + 2 mg CMS. The study started with three sentinel participants receiving either VaxigripTetra (3 µg) + 2 mg CMS or control VaxigripTetra with a 2:1 randomisation. After a 72 h safety review, an additional group of six participants were vaccinated with a 2:1 randomisation. The remaining 21 participants were vaccinated (2:1 randomisation) after another 72 h safety assessment. This study was designed as a dose-escalation study, with a planned two-fold escalation from 1 mg to 4 mg CMS. However, due to an error in calculating the adjuvant concentration (not accounting for the dead volume of the needle), it was discovered that the first group received 2 mg of CMS. A Data Safety Monitoring Board (DSMB) reviewed the safety data from the first 7 days post-vaccination for all 30 participants in this group. Following this review, there was a consensus to reduce the dose and proceed with a two-fold dose reduction of 0.5 mg of CMS. The same procedures were repeated.

Following vaccination, safety monitoring included reporting of solicited and unsolicited adverse events (AE), serious adverse events (SAE), adverse events of special interest (AESIs), and potential immune-mediated events (pIMDs), along with physical examinations, vital signs, and clinical safety laboratory tests. A complete list of all clinical safety parameters tested is provided in Appendix A. Participants utilised an electronic diary for 7 days after study vaccination to record solicited local and systemic AEs, as well as oral body temperature. Solicited local AEs were redness, swelling, induration, and pain at the injection site. Solicited systemic AEs were headache, fatigue, malaise, arthralgia, myalgia, and fever. All reported AEs were evaluated by the investigator and graded according to the “Guidance for Industry: Toxicity Grading Scale for Healthy Adult and Adolescent Volunteers Enrolled in Preventive Vaccine Clinical Trials, Sept. 2007” [12]. Participants were instructed to report any changes in health status until 28 days after vaccination (unsolicited AEs). SAEs, AESIs, and pIMDs were documented until the end of the study. AESIs included anaphylaxis and convulsions, while pIMDs encompassed autoimmune diseases and other inflammatory and/or neurologic disorders of interest which may or may not have an autoimmune aetiology.

Serum samples were collected from all participants at baseline (day 0) and at day 7, day 28, and day 180 for determination of haemagglutination inhibition (HI) titres and microneutralisation (MN) antibody titres against the four influenza virus vaccine strains. Blood samples for peripheral blood mononuclear cell (PBMC) isolation and evaluation of cell-mediated immunity using intracellular cytokine staining were collected at baseline (day 0) and at day 7, day 28, and day 180. Detailed methods for immunological assays are provided in Appendix A.

### 2.4. Endpoints

Primary endpoints included the number of participants experiencing at least one solicited AE during the 7-day post-vaccination period (local, systemic, or either) of any severity grade (mild, moderate, severe), the occurrence of unsolicited AEs during the 28-day post-vaccination period, and the presence of SAEs, AESIs, and pIMDs, in each cohort.

Secondary endpoints were HI and MN antibody titres against the four influenza strains included in VaxigripTetra, assessed in serum samples collected from participants at each timepoint. Seroprotection and seroconversion rates were calculated based on HI titres for all participants. Seroprotection was defined as participants with an HI antibody titre ≥40, while seroconversion was characterised by a four-fold increase in HI antibody titre compared to pre-vaccination level.

Cell-mediated immune responses, defined by the frequency of influenza-specific CD4+ and CD8+ T cells producing at least CD40L, IL-2, IFNγ, and/or TNFα were considered as an exploratory endpoint.

### 2.5. Statistical Analysis

Statistical analyses did not entail formal comparisons between groups; rather, descriptive statistics were used. Discrete variables were summarised, using numbers and percentages, while continuous variables were presented as means (arithmetic or geometric), medians, standard deviations (SD), interquartile ranges (IQR), 95% confidence intervals (CI), minimum (min), and maximum (max). Three analysis sets were defined: the Entered Analysis Set (EAS) comprised all participants who signed the informed consent form, the Safety Analysis Set (SAS) included all participants who received vaccination, and the Immunogenicity Analysis Set (IAS) comprised all participants in the SAS with at least one post-vaccination blood sample analysed for immunogenicity. The IAS and SAS were identical, and results reported here regarding safety and humoral and cell-mediated immunogenicity represent the SAS/IAS. To confirm similar age distribution between cohorts, Kruskal–Wallis test followed by multiple comparisons with Benjamini and Hochberg correction was used. Data analyses, summaries, and listings were generated using SAS version 9.4 M8 (SAS Institute, Inc., Cary, NC, USA) and visualised using GraphPad Prism version 9 (GraphPad Software, San Diego, CA, USA).

## 3. Results

### 3.1. Study Population Demographics

In total, 96 participants were screened for the study, of which 60 (62.5%) were randomised. Seventeen (47.2%) participants did not meet the screening criteria. Others were not assigned to any treatment cohort because the maximum allowable number of 60 participants was reached. Figure 1 provides an overview of the participant allocation across the three cohorts. The vast majority of randomised participants (58 out of 60; 96.7%) completed the study. One participant (5.0%) each of cohort 1 and cohort 2 was lost to follow-up at day 180.

Table 1 presents baseline characteristics and demographics of the study population stratified by cohort. On average, study participants were 37.6 (±10.1) years old, and age was not significantly different between cohorts (*p* = 0.1697). The majority were female (42/60, 70.0%) and white (57/60, 95.0%). The distribution of age and sex was similar across the three cohorts. Mean weight at enrolment was 72.3 (±11.91) kg, and mean BMI was 24.2 (±3.03) kg/m^2^. Among female participants, the vast majority (39/42) were of childbearing potential, representing 92.9% of female participants. None of the participants were pregnant at the time of enrolment.

### 3.2. Safety and Reactogenicity

Figure 2 presents an overview of the number of participants experiencing at least one solicited AE (local and systemic), unsolicited AE, or SAE.

Figure 3 shows the number of participants with at least one solicited local or systemic AE in the 7 days post-vaccination per grade. Solicited local AEs were reported by 57 out of 60 participants (88.3%). Most AEs in all cohorts were mild or moderate in severity and resolved spontaneously within seven days. The most frequently reported solicited local AE was pain (52 out of 60, 86.7%), with the highest frequency reported in participants receiving VaxigripTetra + 2 mg CMS (20 out of 20, 100%). Severe pain at injection site was reported by one participant receiving VaxigripTetra + 2 mg CMS. Solicited systemic AEs were reported by 43 out of 60 participants (71.7%) and were mostly mild or moderate in severity. Overall, the most frequently reported solicited systemic AEs were headache (34 out of 60, 56.7%) and fatigue (33 out of 60, 55.0%), with the highest frequency reported in participants receiving VaxigripTetra + 2 mg CMS (15 out of 20, 75.0% for both AEs). In total, three severe systemic AEs (fatigue, malaise and myalgia) were reported by two participants receiving VaxigripTetra + 2 mg CMS. The majority of solicited systemic AEs were considered related to the study vaccine by the investigator (39 out of 60, 65.0%).

In total, 36 out of 60 participants (60.0%) reported unsolicited AEs. Severe unsolicited AEs were reported by 6 out of 60 participants (10.0%). Among the participants, 21 out of 60 (35.0%) experienced unsolicited AEs considered related to the study vaccine by the investigator. Injection site pain and induration with an onset around day 14 were the most frequently reported unsolicited AEs (8 out of 60, 13.3%, and 4 out of 60, 6.7%, respectively). Induration was typically moderate in severity, with a size of 5.1 cm to 10 cm. It was accompanied by mild pain at the site but without systemic symptoms. All indurations resolved spontaneously around day 28. These were all reported by participants who received a study vaccine with CMS and were considered related to the study vaccine by the investigator.

During the study, one SAE (limb injury) and one pIMD (vestibular neuronitis) were reported. Both were considered unrelated to the study vaccine by the investigator. No fatalities, AEs leading to study discontinuation, or AESIs were reported during the study. There were no clinically significant (CS) changes in actual values or changes from baseline for any haematology or biochemistry clinical laboratory parameter in any cohort within 180 days after study vaccination and results were comparable across cohorts.

### 3.3. Humoral Immune Response

Table 2 shows the geometric mean (95% CI) of HI titres and geometric mean ratio (95% CI) against the four vaccine strains at all timepoints. Figure 4 shows geometric mean (95% CI) HI titres against the four vaccine strains at all timepoints. At baseline, HI GMTs were high but similar for all participants across three cohorts. Across the influenza vaccine strains of VaxigripTetra (A/Darwin (H3N2), A/Victoria (H1N1), B/Austria, and B/Phuket), the HI GMTs increased from baseline to day 7, peaked at day 28, and declined by day 180, although they remained higher compared to baseline.

Overall, HI titres were consistent across strains for the three cohorts at all visits. Participants receiving VaxigripTetra + 2 mg CMS exhibited slightly higher HI GMTs for A/Darwin (H3N2), A/Victoria (H1N1), and B/Phuket compared to the control vaccine. However, for B/Austria, HI GMTs were higher in participants receiving the control vaccine.

Similarly, seroprotection and seroconversion rates were higher on day 28 than on day 180 (Table 2). Seroprotection rates were similar for all cohorts but were higher for A/Darwin and A/Victoria than for influenza B strains. Seroconversion rates were low for all cohorts and influenza strains.

Microneutralisation antibody GMTs showed similar trajectories as HI titres (Appendix A). Again, at baseline, MN GMTs were high but similar across cohorts, increased on day 7, with a peak on day 28, and decreased again to a level higher than baseline on day 180, for all four influenza strains. Participants receiving VaxigripTetra + 2 mg CMS had slightly higher MN GMTs for A/Darwin (H3N2), A/Victoria (H1N1), and B/Phuket, but lower for B/Austria, although MN GMTs were comparable across cohorts.

### 3.4. Cell-Mediated Immune Response

The number of influenza-specific CD4+ polypositive T cells, defined as CD4+ T cells producing at least two of the following immune markers: CD40L, IFNγ, IL-2, and TNFα, exhibited similar trends between cohorts at any visit for any vaccine strain (Figure 5). Generally, CD4+ polypositive T cells followed patterns akin to those observed for HI titres, with an increase from baseline to day 7 and day 28. Although the frequency of CD4+ polypositive T cells declined again by day 180, it did not return to baseline levels. Furthermore, no notable differences were observed between CD4+ T cells producing only one of the markers (CD40L, IFNγ, IL-2, or TNFα), albeit CD4+ T cells producing only CD40L and CD4+ T cells producing only TNFα exhibited the greatest increase across all cohorts post-vaccination. Conversely, no significant differences were detected in the frequencies of influenza-specific CD8+ polypositive T cells between cohorts. However, mean frequencies across all visits for all cohorts remained very low (Appendix A).

## 4. Discussion

In this study, we assessed the safety and immunogenicity of CMS, a novel oil-in-water adjuvant, in combination with a low dose of a licensed quadrivalent split-virion inactivated influenza vaccine. Preclinical investigations demonstrated the tolerability of CMS at doses up to 8 mg (internal report), prompting us to set a conservative maximal dose for the first in-human trial. In two dose levels tested, CMS exhibited safety for human administration. However, participants receiving 2 mg CMS experienced higher reactogenicity compared to those receiving 0.5 mg CMS or the control vaccine. Additionally, delayed onset of pain and induration at the administration site was noted in this group.

Preclinical studies in animal models demonstrated an encouraging safety profile of CMS. In acute and repeated-dose GLP toxicity studies using TETRALITE, 6 µg of HA per influenza virus strain and 8 mg of CMS were administered to both male and female rabbits. No systemic adverse reactions were observed in the acute toxicity study or following the first, second, or third IM injections, except for a transient increase in body temperature ranging from 0.4 and 0.8 °C, peaking at 6 or 24 h post-treatment. The vaccine formulations were well tolerated, with histopathological examination of the injection sites showing minimal to moderate inflammation characterised by the presence of monocytes, lymphocytes, and macrophages in skeletal muscle and adipose tissue.

The adjuvant formulation under investigation bears similarities to two other oil-in-water adjuvants: MF59 and AS03. MF59 is incorporated into the Fluad^®^ vaccine (Seqirus, Maidenhead, UK), recommended for individuals aged 65 and older. AS03 is the adjuvant in the Pandemrix^®^ vaccine (GSK, Rixensart, Belgium) deployed during the H1N1 2009 influenza pandemic. When compared with an AS03-adjuvanted influenza vaccine, similar adverse events and frequencies were observed, primarily including pain, fatigue, and headache [13]. Notably, low-dose VaxigripTetra with 0.5 mg of CMS exhibited a comparable adverse event profile compared to the full dose VaxigripTetra vaccine, while the low-dose (1/5th) VaxigripTetra version with 2.0 mg of CMS demonstrated increased reactivity. Importantly, a delayed local reaction of induration and pain was observed in a considerable number of participants. These delayed reactions have been reported for mRNA vaccines [14] and for adjuvanted vaccines [15,16,17]. While research is sparse, subcutaneous nodules and delayed reactions have been reported for adjuvant systems that exert a depot effect and retain antigen at the injection site for a longer time, more specifically aluminium salts, oil emulsions, and liposomes [17,18]. Delayed reactions are often defined as having an onset later than 72 h post-vaccination with an adjuvant [17], which is significantly earlier than the onset around day 14 post-vaccination for the reactions observed with CMS.

Both humoral and cell-mediated immune responses were comparable between adjuvanted vaccines, even at 1/5th of the antigen dose, and the full-dose control vaccine. The five-fold diluted, CMS-adjuvanted VaxigripTetra generated similar HI and MN antibody titres as the full dose of VaxigripTetra without adjuvant. Thus, CMS stimulates the immune system so that similar responses are obtained with a lower dose of antigen. CMS was able to elicit comparable levels of HI titres and CD4+ polypositive T cells while significantly reducing the antigen dosage.

Indeed, studies suggest that half antigen doses, with or without adjuvant, may be as immunogenic as full doses in a population of younger adults, as seen in this study for CMS [19,20,21]. However, other research indicates a dose-response for doses lower than half for unadjuvanted influenza vaccines, suggesting some limitations to dose-reduction [13]. Several studies comparing different adjuvants in combination with lower dose antigens, have demonstrated an added benefit in elevating HI titres to levels comparable to those achieved by full-dose influenza vaccines [22,23]. For instance, a dose-ranging study with the comparable MF59 adjuvant showed seroconversion rates at day 22 post-vaccination in young to middle-aged adults ranging from 70–91%, with comparability across all groups, even in those receiving the lowest antigen dose (3.75 µg) combined with a half-dose MF59 [21].

Furthermore, there is growing interest in the significance of CD4+ and CD8+ T cell responses following influenza infection and vaccination. It has been established that both live-attenuated and inactivated influenza vaccines can induce a robust antigen-specific CD4+ T cell response, whereas the induction of CD8+ T cell responses tends to be more pronounced after infection and challenging to elicit with inactivated influenza vaccines [24]. Moreover, adjuvants and combinations thereof have been shown to synergistically enhance T-helper 1 polarisation [25,26]. Consistent with our findings, studies have revealed that influenza-specific CD4+ T cells, but not CD8+ T cells, were discernible at day 8 post-vaccination with an MF59-adjuvanted trivalent influenza vaccine, with the quality of the response, as indicated by cytokine production, being comparable to that of an unadjuvanted influenza vaccine [27]. Although the fold change of influenza specific CD4+ T cells post-vaccination inversely correlates with the baseline level, research demonstrates the ability of adjuvants to boost the T cell response even in older adults [28] and to induce distinct transcriptomic signatures [29]. However, one study suggests that optimal CD4+ memory responses necessitate high levels of antigen for non-live vaccines [30]. Intriguingly, the reduction of antigen dose did not seem to impact the cellular immune response when CMS was included.

It is essential to acknowledge that various influenza strains are used across different studies. In our study, the antigens comprised four seasonal influenza strains. Notably, the relatively high HI titres observed on day 0 in our population indicate substantial pre-existing immunity, which may impede the total and relative change in HI titres post-vaccination. In naive populations, such as in studies with low-dose antigens of pandemic influenza strains (as low as 1.9 µg), the adjuvant effect is slightly more pronounced, leading to elevated HI titres above those of unadjuvanted vaccines [31]. However, very low doses still show similar HI titres and seroconversion rates after a single dose, suggesting that antigen dose is more important than pre-existing immunity [13,22,32].

Overall, while reducing antigen doses of influenza vaccines with a factor 5 significantly decreases HI titres, CMS demonstrates comparable effects to other adjuvants and can restore HI titre levels to those of adjuvanted and full-dose unadjuvanted influenza vaccines. The potential of dose-sparing without compromising protective efficacy in younger adults could prove advantageous in situations of limited antigen availability. However, in older adults, where reactogenicity and immunogenicity of seasonal influenza vaccines tend to be lower, a full dose of the antigen combined with a potent adjuvant like CMS remains necessary to induce robust responses and enhance vaccine effectiveness in this population [33,34,35].

The study design has several limitations. First, there is no control group receiving a low, unadjuvanted dose of VaxigripTetra, making it difficult to draw definitive conclusions about the adjuvant effect of CMS. However, since existing literature shows a dose-response relationship with influenza vaccines, particularly when doses are reduced by more than half the standard dose, it is likely that a 1/5th dose of VaxigripTetra without an adjuvant would have lower immunogenicity. Therefore, it is reasonable to assume that CMS contributes to increasing HI titres. Future research with comparisons of equal doses with and without CMS will be needed to confirm this. Second, influenza surveillance was not conducted during the study. Although there are no indications of such occurrences, potential immunological interference from an infection cannot be entirely ruled out. Possible respiratory infections and influenza-like illnesses were documented as unsolicited AEs up to 28 days post-vaccination. However, all participants were vaccinated within the recommended timeframe and prior to the onset of the 2022–2023 epidemic. Therefore, it is likely that infections had a limited impact on the observed HI titres, especially up to day 28. Lastly, although cell-mediated immunity was assessed, the focus was only on a Th1 response. While CMS did not affect the frequency of CD4+ polypositive T cells based on the markers CD40L, IFNγ, IL-2, and TNFα, the inclusion of additional markers might identify other T cell subsets, such as Th2. Other cell types could be differentially activated by CMS, which might provide a better understanding of the mechanisms of action and effect of CMS on increasing immunogenicity under dose-sparing conditions.

In conclusion, CMS exhibits acceptable safety and reactogenicity profiles when combined with a low-dose influenza vaccine in healthy younger adults. Doses exceeding 2 mg are not recommended due to dose-dependent reactogenicity. CMS demonstrates the capacity to enhance the influenza-specific immune response while utilising only 1/5th of the antigen dose, achieving levels comparable to those of a full-dose unadjuvanted influenza vaccine. While increased effectiveness with a reduced antigen dose for older adults may not be achieved, this might be possible when CMS is combined with a full antigen dose, which is investigated in an ongoing study. Moreover, CMS holds potential beneficial implications in low-resource settings or during a pandemic context.

## Figures and Tables

**Figure 1 vaccines-12-01036-f001:**
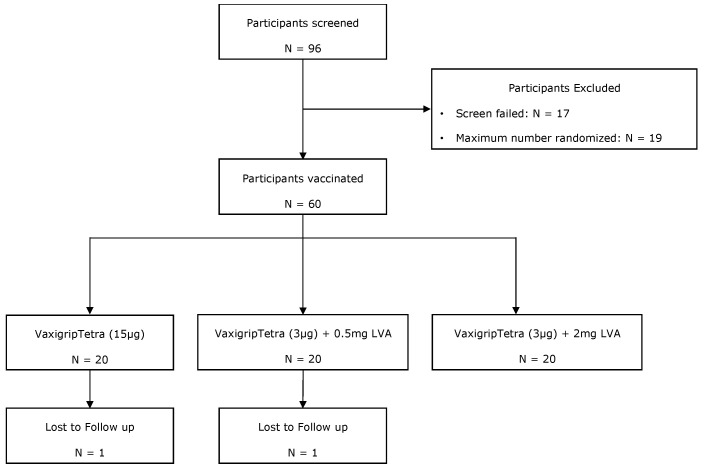
Flowchart of participant recruitment, inclusion, and randomisation.

**Figure 2 vaccines-12-01036-f002:**
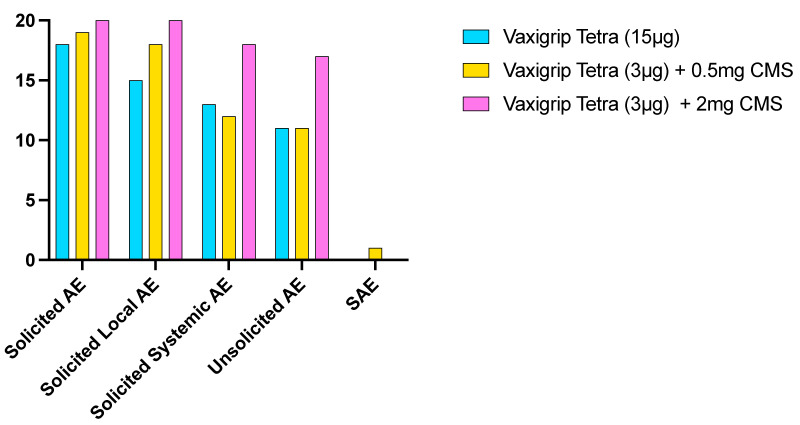
Number of participants per cohort reporting at least one adverse event during the reporting period.

**Figure 3 vaccines-12-01036-f003:**
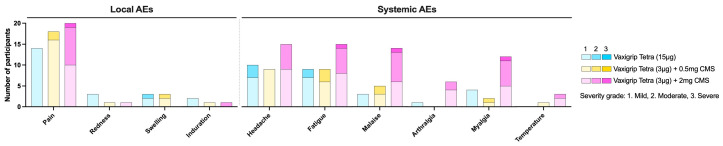
Number of participants per cohort reporting local or systemic adverse events for at least one day of mild, moderate, or severe intensity.

**Figure 4 vaccines-12-01036-f004:**
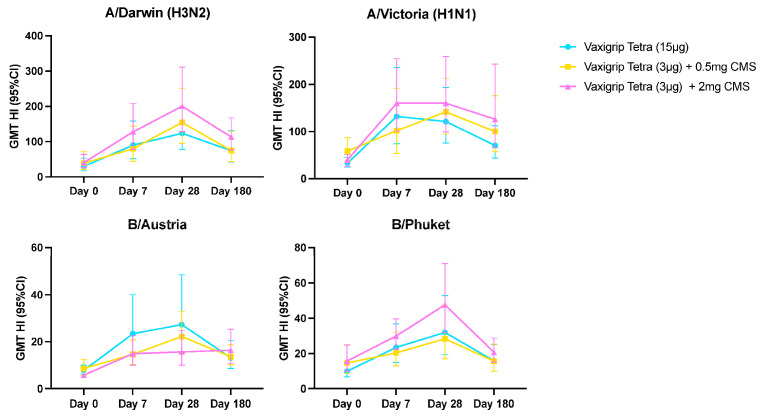
Geometric mean haemagglutination inhibition titres (GMTs) per cohort for each vaccine strain at baseline and 7, 28, and 180 days after vaccination.

**Figure 5 vaccines-12-01036-f005:**
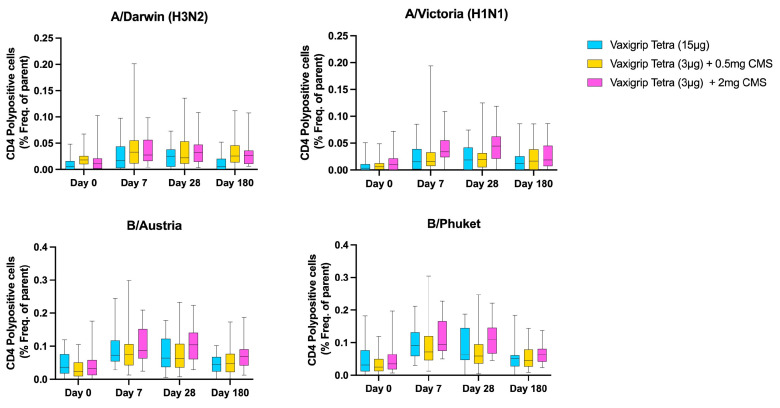
Frequency of CD4+ polypositive T cells per cohort per vaccine strain at baseline and 7, 28, and 180 days after vaccination.

**Table 1 vaccines-12-01036-t001:** Demographic characteristics of randomised and vaccinated participants.

Parameter	Category	VaxigripTetra (15 µg) (N = 20)	VaxigripTetra (3 µg) + 0.5 mg CMS (N = 20)	VaxigripTetra (3 µg) + 2 mg CMS (N = 20)	Total (N = 60)
Age (years (SD))		39.8 (9.8)	38.9 (10.0)	34.1 (10.0)	37.6 (10.1)
Sex (n, %)	Female	12 (60.0%)	14 (70.0%)	16 (80.0%)	42 (70.0%)
	Male	8 (40.0%)	6 (30.0%)	4 (20.0%)	18 (30.0%)
Age and sex distribution (n,%)	18–25 y—Male	2 (10.0%)	0 (0.0%)	0 (0.0%)	2 (3.3%)
	18–25 y—Female	1 (5.0%)	3 (15.0%)	6 (30.0%)	10 (16.7%)
	26–35 y—Male	1 (5.0%)	1 (5.0%)	1 (5.0%)	3 (5.0%)
	26–35 y—Female	1 (5.0%)	4 (20.0%)	4 (20.0%)	9 (15.0%)
	36–50 y—Male	5 (25.0%)	5 (25.0%)	3 (15.0%)	13 (21.7%)
	36–50 y—Female	10 (50.0%)	7 (35.0%)	6 (30.0%)	23 (38.3%)
Race (n, %)	White	20 (100.0%)	19 (95.0%)	18 (90.0%)	57 (95.0%)
	Black or African American	0 (0.0%)	1 (5.0%)	0 (0.0%)	1 (1.7%)
	Asian	0 (0.0%)	0 (0.0%)	1 (5.0%)	1 (1.7%)
	Other	0 (0.0%)	0 (0.0%)	1 (5.0%)	1 (1.7%)
Weight (kg (SD))		74.34 (13.4)	73.96 (12.78)	68.49 (8.68)	72.26 (11.91)
BMI (kg/m^2^ (SD))		24.33 (3.43)	24.33 (3.07)	23.84 (2.69)	24.17 (3.03)

BMI: body mass index, SD: standard deviation; CMS: carbohydrate fatty acid monosulphate esters.

**Table 2 vaccines-12-01036-t002:** Geometric mean haemagglutination inhibition titres (GMTs), seroconversion, and seroprotection rates per cohort per vaccine strain at baseline and 7, 28, and 180 days after vaccination.

		VaxigripTetra (15 µg)				VaxigripTetra (3 µg) + 0.5 mg CMS				VaxigripTetra (3 µg) + 2 mg CMS			
		A/Darwin (H3N2)	A/Victoria (H1N1)	B/Austria	B/Phuket	A/Darwin (H3N2)	A/Victoria (H1N1)	B/Austria	B/Phuket	A/Darwin (H3N2)	A/Victoria (H1N1)	B/Austria	B/Phuket
**Day 0**	GMT (95% CI)	**30.3** (17.3–53.1)	**33.1** (24.4–44.8)	**7.8** (6.2–9.9)	**9.8** (6.7–14.4)	**39.3** (21.6–71.7)	**58.6** (39.5–86.8)	**8.6** (5.8–12.5)	**14.6** (8.7–24.7)	**40.7** (25.8–64.3)	**40** (25.9–61.7)	**5.7** (4.9–6.7)	**15.7** (9.9–24.8)
**Day 7**	GMT (95% CI)	**90.3** (51.5–158.5)	**132.2** (74.0–236.2)	**23.4** (13.7–40.0)	**23.4** (14.8–36.9)	**80.0** (44.3–144.4)	**102.0** (54.2–191.8)	**14.6** (10.3–20.8)	**20.3** (12.9–32.1)	**127.7** (78.6–207.7)	**160.0** (100.5–254.7)	**14.9** (9.9–22.5)	**29.8** (22.4–39.6)
	GMR (95% CI)	3.0 (1.9–4.6)	4.0 (2.2–7.4)	3.0 (1.9–4.7)	2.4 (1.5–3.7)	2.0 (1.5–2.8)	1.7 (1.1–2.7)	1.7 (1.2–2.4)	1.4 (1.0–1.9)	3.1 (2.1–4.6)	4.0 (2.7–6.0)	2.6 (1.8–3.7)	1.9 (1.2–3.0)
**Day 28**	GMT (95% CI)	**123.4** (78.5–193.9)	**121.3** (75.8–193.9)	**27.3** (15.4–48.5)	**31.9** (19.3–52.9)	**154.5** (95.2–251.0)	**141.7** (94.2–213.2)	**22.2** (15.0–32.9)	**28.3** (16.9–47.4)	**200.4** (128.9–311.6)	**160.0** (98.8–259.2)	**15.7** (9.9–24.8)	**47.6** (31.9–70.9)
	GMR (95% CI)	4.1 (2.4–7.0)	3.7 (2.1–6.5)	3.5 (2.1–5.8)	3.2 (1.9–5.7)	3.9 (2.5–6.1)	2.4 (1.6–3.6)	2.6 (1.8–3.7)	1.9 (1.3–2.9)	4.9 (3.3–7.4)	4.0 (2.6–6.2)	2.7 (1.8–4.0)	3.0 (1.9–4.9)
	SPR (n, %)	19 (95.0%)	20 (100.0%)	8 (40.0%)	11 (55.0%)	19 (95.0%)	19 (95.0%)	7 (35.0%)	11 (55.0%)	20 (100.0%)	20 (100.0%)	3 (15.0%)	15 (75.0%)
	SCR (n, %)	9 (45.0%)	9 (45.0%)	8 (40.0%)	8 (40.0%)	9 (45.0%)	7 (35.0%)	6 (30.0%)	5 (25.0%)	14 (70.0%)	8 (40.0%)	6 (30.0%)	9 (45.0%)
**Day 180**	GMT (95% CI)	**74.4** (42.2–131.1)	**70.4** (44.0–112.6)	**13.1** (8.5–20.4)	**15.8** (10.0–25.0)	**74.6** (43.1–129.2)	**100.2** (57.1–175.9)	**13.7** (10.1–18.6)	**15.7** (9.8–25.2)	**113.1** (76.6–167.1)	**126.2** (65.5–243.4)	**16.4** (10.6–25.3)	**20.7** (15.0–28.7)
	GMR (95% CI)	2.2 (1.4–3.6)	2.0 (1.2–3.3)	1.6 (1.1–2.4)	1.6 (0.9–2.7)	1.9 (1.2–2.9)	1.7 (1.1–2.8)	1.6 (1.1–2.4)	1.1 (0.8–1.5)	3.0 (2.0–4.5)	3.1 (1.8–5.3)	2.8 (1.9–4.2)	1.3 (0.8–2.0)
	SPR (n, %)	15 (75.0%)	16 (80.0%)	2 (10.0%)	2 (10.0%)	14 (70.0%)	18 (90.0%)	3 (15.0%)	4 (20.0%)	18 (90.0%)	16 (80.0%)	4 (20.0%)	5 (25.0%)
	SCR (n, %)	4 (20.0%)	5 (25.0%)	3 (15.0%)	6 (30.0%)	5 (25.0%)	5 (25.0%)	4 (20.0%)	2 (10.0%)	8 (40.0%)	6 (30.0%)	7 (35.0%)	4 (20.0%)

GMT: geometric mean titre; GMR: geometric mean ratio compared to day 0; SPR: seroprotection rate; SCR: seroconversion rate; 95% CI: 95% confidence interval.

## Data Availability

Raw data, including deidentified participant data, will be made available upon reasonable request to the corresponding author.

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
