# Peer review of "Safety and Immunogenicity of a Carbohydrate Fatty Acid Monosulphate Ester Adjuvant Combined with a Low-Dose Quadrivalent Split-Virion Inactivated Influenza Vaccine: A Randomised, Observer-Blind, Active-Controlled, First-in-Human, Phase 1 Study"

_vaccines, 2024, doi:10.3390/vaccines12091036_

Round 1
Reviewer 1 Report
Comments and Suggestions for Authors
This study evaluates the safety and immunogenicity of CMS when added to a low dose influenza vaccine (QIV) in humans. The use of this adjuvant reduces the cost and adverse reactions of influenza virus vaccine, which is conducive to the prevention of influenza virus. However, there are some problems in this manuscript. There are my suggestions.
1. In line 60 of the manuscript, authors mention that CMS can be used to improve vaccination effectiveness in older adults. However, there is no relevant research cited. It needs to be supplemented in the introduction.
2. The background of the influenza vaccine (VaxigripTetra) used in this study is not detailed and needs to be added in the introduction.
3. The healthy control group and a single low-dose vaccine group were lacking in all experimental results, and these data must be supplemented in the manuscript.
4. Were all participants tested for influenza virus infection prior to vaccination? This is necessary for clinical experiments of influenza vaccines. These data must be supplemented in the manuscript.
5. Whether the participants blood routine examination and blood physiological and biochemical indexes were tested after immunization?These data are important indicator of vaccine safety and must be supplemented in the manuscript.
6. The participants in VaxigripTetra (3µg) +2 mg CMS group were younger than the other groups. Does this factor affect the results of the study? These need to be added in the discussion.
7. Is reported the mouse and human safety test of CMS in past research? Is this experiment based on the existing safety data? These need to be added in the discussion.
Author Response
This study evaluates the safety and immunogenicity of CMS when added to a low dose influenza vaccine (QIV) in humans. The use of this adjuvant reduces the cost and adverse reactions of influenza virus vaccine, which is conducive to the prevention of influenza virus. However, there are some problems in this manuscript. There are my suggestions.
We thank the reviewer for their time and insightful suggestions, which have greatly improved the manuscript.
- In line 60 of the manuscript, authors mention that CMS can be used to improve vaccination effectiveness in older adults. However, there is no relevant research cited. It needs to be supplemented in the introduction.
Since this is the first time CMS has been tested in humans, there is until today no data or research available to support improved effectiveness in older adults by using CMS. Therefore, the sentence in the introduction has been changed to reflect the still hypothetical nature of the possible benefits for future use of CMS in older adults.
The sentence now reads as follows:
Consequently, CMS may provide benefits like dose-sparing and improved immunogenicity when administered to older adults
- The background of the influenza vaccine (VaxigripTetra) used in this study is not detailed and needs to be added in the introduction.
The following information on VaxigripTetra has been added to the introduction, Line 39-45:
One widely used vaccine is VaxigripTetra®, a quadrivalent inactivated split virion influenza virus vaccine. The 2022-2023 seasonal vaccine contains 15µg HA of each of the two influenza type A virus subtypes (A/Darwin (H3N2) and A/Victoria (H1N1)) and the two influenza B virus subtypes (B/Austria, and B/Phuket), compliant with the WHO recommendations for the 2022/23 season. This vaccine is known to induce superior immunogenicity compared to trivalent inactivated influenza vaccines as assessed by HI titres.
- The healthy control group and a single low-dose vaccine group were lacking in all experimental results, and these data must be supplemented in the manuscript.
Indeed, there are no results of a single low-dose vaccine group. This group has not been included in the study. Thus, these results are not available and cannot be added here. This is one of the limitations of the study that was already discussed in lines 498-594 and has been edited to further clarify these issues.
We are not sure what exactly the reviewer means when referring to “a healthy control group”. In this study, all included and vaccinated participants are healthy adults aged between 18 and 50 years. We feel that the study is sufficiently controlled by using a group that receives the standard of care, namely VaxigripTetra for the 2022-2023 season. Because a licensed vaccine is available, no group placebo group was included as this would breach ethical guidelines. Additionally, comparing safety and immunogenicity against a placebo does not adequately reflect real-world practice.
- Were all participants tested for influenza virus infection prior to vaccination? This is necessary for clinical experiments of influenza vaccines. These data must be supplemented in the manuscript.
None of the participants were tested for influenza virus infection prior to vaccination and this data cannot be provided. However, we dare to disagree with the reviewer that this information is necessary for clinical studies. Testing for influenza could indeed explain unusually high baseline HAI titres; however, such testing only identifies acute infection shortly before vaccination and fails to provide information about infections that occurred in the months prior to vaccination. To reduce the possibility of prior infection, several strategies were adopted. First, individuals who had a confirmed influenza infection within the 12 months prior to vaccination were excluded from the study. Second, vaccinations took place in October-November 2022, aligning with the recommended influenza vaccination period in Belgium, several months before the start of the seasonal epidemic when virus circulation is minimal. Finally, all participants were confirmed to be healthy at time of enrolment and vaccination, with no indications of respiratory infection.
The method section, lines 103-109, has been rewritten to further clarify this issue:
Participants in the study were healthy male and female adults aged 18 and 50 years who had not had a confirmed influenza virus infection in the 12 months preceding study vaccination and had not received an influenza vaccination in the preceding 6 months. To further minimize the likelihood of unusually high baseline HI titres, vaccinations were administered in October-November 2022, the recommended influenza vaccination period in Belgium, before the onset of the seasonal epidemic when virus circulation is low.
- Whether the participants blood routine examination and blood physiological and biochemical indexes were tested after immunization? These data are important indicator of vaccine safety and must be supplemented in the manuscript.
As noted in Line 126 of the original submission, blood safety parameters were assessed at all study visits. The following parameters were evaluated: haemoglobin, haematocrit, platelet count, red blood cell (RBC) count, mean corpuscular volume, mean corpuscular haemoglobin, reticulocytes, white blood cell (WBC) count, absolute neutrophil count, absolute lymphocyte count, absolute monocyte
count, absolute eosinophil count, and absolute basophil count. urea, creatinine, aspartate aminotransferase (AST), alanine aminotransferase (ALT), alkaline phosphatase, total and direct bilirubin, C-reactive protein, glucose (nonfasted), and haemoglobin A1c. A listing of these tests is now provided in the Supplementary Table 2.
There were no clinically significant changes from baseline for tested safety parameters and results were comparable across the 3 cohorts.
This has been added in the Results section on Line 323-325:
There were no clinically significant (CS) changes in actual values or changes from baseline for any haematology or biochemistry clinical laboratory parameter in any cohort within 180 days after study vaccination and results were comparable across cohorts.
- The participants in VaxigripTetra (3µg) +2 mg CMS group were younger than the other groups. Does this factor affect the results of the study? These need to be added in the discussion.
Age is indeed a factor that can influence the magnitude of the immune response and the severity of adverse events. However, we do not believe it plays a significant role in this study as a well-defined age group (18-50 years of age) has been selected. Within this age range, the variability in immunogenicity is limited. Additionally, we can confirm that there were no significantly age differences across groups, and this information has been added to the text. To provide further clarification, Table 1, which displays demographic characteristics, has been updated to include a more detailed age distribution. The distribution of sex and age across the three cohorts was also similar, with no significant different between cohorts. This was assessed using Kruskal-Wallis test followed by multiple comparisons with Benjamini and Hochberg correction.
Methods Line 258-260:
To confirm similar age distribution between cohorts, Kruskal-Wallis test followed by multiple comparisons with Benjamini and Hochberg correction was used.
Results Line 275-278:
On average, study participants were 37.6 (±10.1) years old, and age was not significantly different between cohorts (p = 0.1697). The majority were female (42/60, 70.0%) and white (57/60, 95.0%). The distribution of age and sex was similar across the 3 cohorts.
- Is reported the mouse and human safety test of CMS in past research? Is this experiment based on the existing safety data? These need to be added in the discussion.
This manuscript reports the safety data of CMS in humans for the first time, as this is the initial human study conducted with CMS. Pre-clinical animal studies have been performed, and their details have now been added to the discussion, line 407-415.
Preclinical studies in animal models demonstrated an encouraging safety profile of CMS. In acute and repeated-dose GLP toxicity studies using TETRALITE, 6µg of HA per influenza virus strain and 8 mg of CMS were administered to both male and female rabbits. No systemic adverse reactions were observed in the acute toxicity study or following the first, second, or third IM injections, except for a transient increase in body temperature ranging from 0.4 and 0.8 ËšC, peaking at 6 or 24 h post- treatment. The vaccine formulations were well tolerated, with histopathological examination of the injection sites showing minimal to moderate inflammation characterized by the presence of monocytes, lymphocytes and macrophages in skeletal muscle and adipose tissue.
Reviewer 2 Report
Comments and Suggestions for Authors
Comments for the authors of Vaccines manuscript vaccines-3140233:
The author of the Vaccines manuscript “Safety and immunogenicity of a carbohydrate fatty acid monosulphate ester adjuvant combined with a low dose quadrivalent split-virion inactivated influenza vaccine: a randomized, observer-blind, active-controlled, first-in-human, phase 1 study”, present their recent work with the adjuvant CMS. Specifically, the authors add CMS to a low-dose quadrivalent vaccine in a first-in-human trial. There were 3 trial arms with the vaccine delivered as either a full-dose or a 1/5th dose with either 0.5 or 2.0 milligrams CMS as an adjuvant. Patients in the study were 18-50 years of age, and adverse events were directly followed for 7 days post-vaccination. The research team evaluated both antibody (hemagglutination inhibition) and CD4+ T cell responses at 0, 7, 28, and 180 days-post-vaccination. Vaccine recipients reported pain at injection site (86.7%) and headache (56.7%) in the 2 mg CMS group. Antibody and T cell responses increased until Day 28 and then decreased, while remaining above the baseline, at Day 180, demonstrating both safety and efficacy. Below are some comments that I would like the authors to address as they revise the manuscript.
Comments:
- In the abstract, it is not totally clear that the full dose is delivered without adjuvant. Re-wording this sentence might help the reader understand the study design a bit better.
- Since the antibody data are critical for understanding the impact of the adjuvant on vaccine recipients, it would help if these data were represented in a figure and/or if the key titer comparisons were bolded and/or highlighted to show the differences described in the text.
- In Table 2, the columns could also be better aligned with the vaccine groups to make it more clear which heading goes with which vaccine.
- The authors claim that the titers with adjuvant were similar with 1/5th dose than with full dose, but there is not a direct comparison with 1/5th dose without adjuvant. This makes it difficult to understand if the adjuvant is truly having an effect or if 1/5th of the dose would work as well as a full-dose in this target population.
- The authors clearly indicated the three limitations in their study design, and added discussion around these limitations, but the impact of these limitations don’t necessarily alter the conclusions drawn. This is especially problematic with the absence of the 1/5th dose group without adjuvant.
Author Response
The author of the Vaccines manuscript “Safety and immunogenicity of a carbohydrate fatty acid monosulphate ester adjuvant combined with a low dose quadrivalent split-virion inactivated influenza vaccine: a randomized, observer-blind, active-controlled, first-in-human, phase 1 study”, present their recent work with the adjuvant CMS. Specifically, the authors add CMS to a low-dose quadrivalent vaccine in a first-in-human trial. There were 3 trial arms with the vaccine delivered as either a full-dose or a 1/5th dose with either 0.5 or 2.0 milligrams CMS as an adjuvant. Patients in the study were 18-50 years of age, and adverse events were directly followed for 7 days post-vaccination. The research team evaluated both antibody (hemagglutination inhibition) and CD4+ T cell responses at 0, 7, 28, and 180 days-post-vaccination. Vaccine recipients reported pain at injection site (86.7%) and headache (56.7%) in the 2 mg CMS group. Antibody and T cell responses increased until Day 28 and then decreased, while remaining above the baseline, at Day 180, demonstrating both safety and efficacy. Below are some comments that I would like the authors to address as they revise the manuscript.
We thank the reviewer for their feedback and suggestions that improve this manuscript.
Comments:
- In the abstract, it is not totally clear that the full dose is delivered without adjuvant. Re-wording this sentence might help the reader understand the study design a bit better.
The sentence has been reworded as suggested by the reviewer:
In a randomized, double-blind, active-controlled, first-in-human study, sixty participants (18–50 years) received either 0.5mg CMS or 2mg CMS with 1/5th dose QIV, or a full dose QIV without CMS.
- Since the antibody data are critical for understanding the impact of the adjuvant on vaccine recipients, it would help if these data were represented in a figure and/or if the key titer comparisons were bolded and/or highlighted to show the differences described in the text.
- In Table 2, the columns could also be better aligned with the vaccine groups to make it more clear which heading goes with which vaccine.
A figure (Figure 4) has been added to show the geometric mean haemagglutination inhibition titers and 95% CI per cohort for each vaccine strain on all examination days. Since Table 2 provides additional information not captured in the figure, we have decided to keep the table as well. It has been updated to better display and align all important results.
- The authors claim that the titers with adjuvant were similar with 1/5th dose than with full dose, but there is not a direct comparison with 1/5th dose without adjuvant. This makes it difficult to understand if the adjuvant is truly having an effect or if 1/5th of the dose would work as well as a full dose in this target population.
- The authors clearly indicated the three limitations in their study design, and added discussion around these limitations, but the impact of these limitations doesn’t necessarily alter the conclusions drawn. This is especially problematic with the absence of the 1/5th dose group without adjuvant.
We agree that understanding the effect of the adjuvant is challenging without a direct comparison to a 1/5th dose without adjuvant. Therefore, this has been explicitly mentioned as one of the study’s limitations. However, we also discuss the existing literature (beginning on line 453) that demonstrates clear dose-response relationships with influenza vaccines, particularly when doses are reduced by more than half the standard dose. This evidence suggests a definite need for sufficient antigen doses to achieve adequate HI titres. Additionally, the use of high-dose influenza vaccines in older adults to enhance immunogenicity further confirms the dose-effect on of HI titre production. Consequently, it is likely that a 1/5th dose of Vaxigrip without an adjuvant would have lower immunogenicity, and it is reasonable to assume that CMS contributes to increasing HI titres. Future research with comparisons of equal doses with and without CMS will be conducted to explore this further. We have updated this limitation to accurately reflect this reasoning.
The other limitations have been clarified as well.
The study design has several limitations. First, there is no control group receiving a low, unadjuvanted dose of VaxigripTetra, making it difficult to draw definitive conclusions about the adjuvant effect of CMS. However, since existing literature shows a dose-response relationship with influenza vaccines, particularly when doses are reduced by more than half the standard dose, it is likely that a 1/5th dose of VaxigripTetra without an adjuvant would have lower immunogenicity. Therefore, it is reasonable to assume that CMS contributes to increasing HI titres. Future research with comparisons of equal doses with and without CMS will be needed to confirm this. Second, influenza surveillance was not conducted during the study. Although there are no indications of such occurrences, potential immunological interference from an infection cannot be entirely ruled out. Possible respiratory infections and influenza-like illnesses were documented as unsolicited AEs up to 28 days post-vaccination. However, all participants were vaccinated within the recommended timeframe and prior to the onset of the 2022-2023 epidemic. Therefore, it is likely that infections had a limited impact on the observed HI titres, especially up to day 28. Lastly, although cell-mediated immunity was assessed, the focus was only on a Th1 response. While CMS did not affect the frequency of CD4+ polypositive T cells based on the markers CD40L, IFN-g, IL-2, and TNF-a, the inclusion of additional markers might identify other T cell subsets, such as Th2. Other cell types could be differentially activated by CMS, which might provide a better understanding of the mechanisms of action and effect of CMS on increasing immunogenicity under dose-sparing conditions.
Reviewer 3 Report
Comments and Suggestions for Authors
The study entitled on Safety and immunogenicity of a carbohydrate fatty acid monosulphate ester adjuvant combined with a low dose quadrivalent split-virion inactivated influenza vaccine: a randomized, observer-blind, active-controlled, first-in-human, phase 1 study.
Comments :
11. What does the first line of the abstract say: CMS? The abbreviation should be explained, giving the expanded form first, and the abbreviation can be used afterwards.
22. The methodology mentions that healthy male and female adults between the ages of 18 and 50 participated in the study. However, the study procedure states that sixty healthy participants were randomly assigned to three cohorts of 20 people each, in camera. The author should indicate the number of male and female participants and their age distribution, e.g. 18-25; 26-35; and 36-50.
3. Three groups were compared in the study: the full dose of VaxigripTetra (15 µg) as a control vaccine, a low dose of VaxigripTetra (3 µg) in combination with 0.5 mg CMS and another low dose of VaxigripTetra (3 µg) in combination with 2 mg CMS. The CMS doses of 0.5 mg and 2 mg, how these specific amounts were chosen in this study?
43. What impact does the high percentage (92.9%) of female participants of childbearing age have on the study results and safety assessments?
54. Page number 10, Line 328, spelling of first. Why the study has limitation that there is no control group receiving a low unadjuvanted dose of VaxigripTetra, making it challenging to draw conclusions regarding the adjuvant effect of CMS. Second, influenza surveillance was not conducted during the study, and although there are no indications, potential interferences by an infection cannot be ruled out. What is the reason for the limitation.
65. The lack of significant differences in the frequency of influenza-specific CD8+ polypositive T cells between the cohorts, and why do these CD8+ T cell frequencies remain very low throughout the study?
76. The study lacks physicochemical characterization and stability assessment of the adjuvant developed by CMS.
87. Due to numerous typographical errors, the manuscript must be proofread by a professional proofreader.
98. The references cited in the study are not in MDPI format. They should be reformatted to adhere to the MDPI reference style guidelines.
9
Author Response
The study entitled on Safety and immunogenicity of a carbohydrate fatty acid monosulphate ester adjuvant combined with a low dose quadrivalent split-virion inactivated influenza vaccine: a randomized, observer-blind, active-controlled, first-in-human, phase 1 study.
We thank the reviewer for their time and valuable feedback that helped us improve this manuscript
Comments:
- What does the first line of the abstract say: CMS? The abbreviation should be explained, giving the expanded form first, and the abbreviation can be used afterwards.
We thank the reviewer for this comment. The abstract has been changed accordingly.
- The methodology mentions that healthy male and female adults between the ages of 18 and 50 participated in the study. However, the study procedure states that sixty healthy participants were randomly assigned to three cohorts of 20 people each, in camera. The author should indicate the number of male and female participants and their age distribution, e.g. 18-25; 26-35; and 36-50.
The table has been updated to include age and sex distribution. The text has been clarified to indicate that this distribution did not differ between cohorts.
On average, study participants were 37.6 (±10.1) years old, and age was not significantly different between cohorts (p = 0.1697). The majority were female (42/60, 70.0%) and white (57/60, 95.0%). The distribution of age and sex was similar across the 3 cohorts.
- Three groups were compared in the study: the full dose of VaxigripTetra (15 µg) as a control vaccine, a low dose of VaxigripTetra (3 µg) in combination with 0.5 mg CMS and another low dose of VaxigripTetra (3 µg) in combination with 2 mg CMS. The CMS doses of 0.5 mg and 2 mg, how these specific amounts were chosen in this study?
These doses were chosen based on data from preclinical animal studies. In the nonclinical studies with CMS in ferrets, rabbits and pigs, doses of 4 or 8 mg of CMS were used. For the GLP toxicology studies, a decision was made to investigate a dose that was two-fold higher than the maximum intended human dose, specifically 8 mg of CMS. In the acute and repeated-dose toxicity studies in rabbits, a two-fold higher dose of TETRALITE was investigated, which included 6 µg of HA and 8 mg of CMS. Based on results from the pivotal immunogenicity and dose-finding study with TETRALITE in ferrets, the maximum intended human dose of LVA was set at 4 mg. The antigen dose was fixed at 3 µg of HA per strain of each of the 4 influenza strains (equivalent to 1/5th of VaxigripTetra®). These doses were selected because they combined a higher immune response with minimal or no adverse effects.
This study was designed as a dose-escalation study, with a planned two-fold escalation from 1mg to 4mg CMS. However, due to an error in calculating the adjuvant concentration (not accounting for the dead volume of the needle), it was discovered that the first group received 2 mg of CMS instead of the intended 1mg. A Data Safety Monitoring Board (DSMB) reviewed the safety data from the first 7 days post-vaccination for all 30 participants in this group. Following this review, there was a consensus to reduce the dose and proceed with a two-fold dose reduction of 0.5 mg of CMS. This has been detailed in the Methods section (line 195-201).
- What impact does the high percentage (92.9%) of female participants of childbearing age have on the study results and safety assessments?
The percentage provided (92.2%) reflects the number of women of childbearing potential (WOCBP) within the female group. While this percentage is high, it is typical and expected within the 18-50 age range. Sex is known to affect vaccine-induced immune responses and reactogenicity, with women generally exhibiting stronger inflammatory responses and higher antibody production post-vaccination compared to men. This effect is less pronounced in postmenopausal women. Consequently, the high proportion of women in this study may lead to increased immunogenicity. However, the distribution of men and women across the cohorts is similar and not significantly different. Therefore, this should not affect the assessment of safety and immunogenicity of Vaxigrip + CMS compared to Vaxigrip alone, nor alter the study’s results or conclusions.
- Page number 10, Line 328, spelling of first. Why the study has limitation that there is no control group receiving a low unadjuvanted dose of VaxigripTetra, making it challenging to draw conclusions regarding the adjuvant effect of CMS. Second, influenza surveillance was not conducted during the study, and although there are no indications, potential interferences by an infection cannot be ruled out. What is the reason for the limitation.
We agree that understanding the effect of the adjuvant is challenging without a direct comparison to a 1/5th dose without adjuvant. Therefore, this has been explicitly mentioned as one of the study’s limitations. However, we also discuss the existing literature (beginning on line 453) that demonstrates clear dose-response relationships with influenza vaccines, particularly when doses are reduced by more than half the standard dose. This evidence suggests a definite need for sufficient antigen doses to achieve adequate HI titres. Additionally, the use of high-dose influenza vaccines in older adults to enhance immunogenicity further confirms the dose-effect on of HI titre production. Consequently, it is likely that a 1/5th dose of Vaxigrip without an adjuvant would have lower immunogenicity, and it is reasonable to assume that CMS contributes to increasing HI titres. Future research with comparisons of equal doses with and without CMS will be conducted to explore this further. We have updated this limitation to accurately reflect this reasoning. As pointed out by the reviewer no influenza surveillance was performed during the study. Both issues are indeed limitations of the study that have been explicitly mentioned in the discussion and phrased as follows:
The study design has several limitations. First, there is no control group receiving a low, unadjuvanted dose of VaxigripTetra, making it difficult to draw definitive conclusions about the adjuvant effect of CMS. However, since existing literature shows a dose-response relationship with influenza vaccines, particularly when doses are reduced by more than half the standard dose, it is likely that a 1/5th dose of VaxigripTetra without an adjuvant would have lower immunogenicity. Therefore, it is reasonable to assume that CMS contributes to increasing HI titres. Future research with comparisons of equal doses with and without CMS will be needed to confirm this. Second, influenza surveillance was not conducted during the study. Although there are no indications of such occurrences, potential immunological interference from an infection cannot be entirely ruled out. Possible respiratory infections and influenza-like illnesses were documented as unsolicited AEs up to 28 days post-vaccination. However, all participants were vaccinated within the recommended timeframe and prior to the onset of the 2022-2023 epidemic. Therefore, it is likely that infections had a limited impact on the observed HI titres, especially up to day 28.
- The lack of significant differences in the frequency of influenza-specific CD8+ polypositive T cells between the cohorts, and why do these CD8+ T cell frequencies remain very low throughout the study?
VaixigripTetra, the vaccine used in this study, is an inactivated split virion vaccine. Protein vaccines like this one are known to induce very weak CD8+ T cell responses. The minimal CD8+ response detected in participants, even at baseline, were likely due to past natural influenza infections. The administration of a split virus vaccine does not increase the frequency of these CD8+ T cells. Although no CD8+ T cell responses were observed, clear CD4+ responses were measurable in the study participants.
- The study lacks physicochemical characterization and stability assessment of the adjuvant developed by CMS.
We do not think this would improve the manuscript. Physiochemical characterization and stability assessment has been performed, and data are available in internal reports. However, the purpose of this manuscript is to report the findings of a clinical trial and its objectives.
- Due to numerous typographical errors, the manuscript must be proofread by a professional proofreader.
Thank you for your valuable feedback on our manuscript. We appreciate your attention to detail. In response to your comment regarding typographical errors, we have conducted a thorough review of the manuscript to ensure accuracy. I believe that the manuscript is now free of significant typographical errors. If there are specific sections that you feel require further attention, we would be grateful for your guidance.
- The references cited in the study are not in MDPI format. They should be reformatted to adhere to the MDPI reference style guidelines.
The references have now been formatted according to the MDPI guidelines using the ACS style guide format.
Reviewer 4 Report
Comments and Suggestions for Authors
Major comments
The authors tested a new adjuvant in combination with QIV. The study is well written. The description of safety data is clear; however the immunogenicity data might not be very conclusive. To know whether the adjuvant has had a real effect in immunogenicity, a group comparing CMS versus oil in water only, and adjuvanted QIV at same dose and a comparator like Alum, MF-59 or AS03 would be required.
Why does CMS produce adverse reactions? Is it the same as Alum? How would CMS behave in the absence of oil in water emulsion?
Have the authors measured Th1/Th2 immune profiles?
How does HI, MNT and T cell immunity correlate within the same participant? Is there also a correlation with side effects?
Minor comments
Line 15. Define CMS
Introduction
Please describe CMS in the introduction.
Methods
How does the final CMS+ Squalene in water emulsion compare to MF-59 formulation? How much of the immunogenicity is due to the emulsion only?
Author Response
We would like to thank the reviewer for their comments that have clearly improved the manuscript.
Major comments
The authors tested a new adjuvant in combination with QIV. The study is well written. The description of safety data is clear; however, the immunogenicity data might not be very conclusive. To know whether the adjuvant has had a real effect in immunogenicity, a group comparing CMS versus oil in water only, and adjuvanted QIV at same dose and a comparator like Alum, MF-59 or AS03 would be required.
Understanding the effect of the adjuvant is indeed challenging without a direct comparison to a 1/5th dose without adjuvant. Therefore, this has been explicitly mentioned as one of the study’s limitations. However, we also discuss the existing literature (beginning on line 453) that demonstrates clear dose-response relationships with influenza vaccines, particularly when doses are reduced by more than half the standard dose. This evidence suggests a definite need for sufficient antigen doses to achieve adequate HI titres. Additionally, the use of high-dose influenza vaccines in older adults to enhance immunogenicity further confirms the dose-effect on of HI titre production. Consequently, it is likely that a 1/5th dose of Vaxigrip without an adjuvant would have lower immunogenicity, and it is reasonable to assume that CMS contributes to increasing HI titres. Future research with comparisons of equal doses with and without CMS will be conducted to explore this further. We have updated this limitation to accurately reflect this reasoning.
We agree with the reviewer that comparing CMS-adjuvanted vaccines with other adjuvanted or high- dose vaccines would be valuable. However, we respectfully disagree that such comparisons are strictly necessary. The primary objective of our study was to compare CMS-adjuvanted vaccines to the standard of care, which is the unadjuvanted QIV used annually for seasonal influenza prevention. While it may be scientifically interesting to compare with other adjuvanted influenza vaccines, using the unadjuvanted vaccine as a comparator is sufficient to evaluate the added value of CMS.
Why does CMS produce adverse reactions? Is it the same as Alum? How would CMS behave in the absence of oil in water emulsion?
CMS is the active ingredient incorporated into a submicron emulsion of squalane in phosphate buffered saline (PBS) and stabilized by Polysorbate 80. Consequently, CMS is similar to other oil-in-water adjuvants, such as MF59 and AS03, which are discussed in the Discussion section.
However, the mechanisms of action are not yet fully understood. The physical interaction, such as the formation of an antigen depot, does not play a major role in the adjuvanticity of CMS. Despite its striking chemical similarities with monophosphoryl lipid A, CMS did not activate Toll-like receptors (TLRs), including TLR4, in vitro. In vivo studies in mice showed an increase in interleukins and cytokines (such as IL-5, IL-6 and especially G-CSF) one day after immunization with influenza virus plus CMS. Recent studies in mice using SARS-COV-2 receptor-binding protein combined with CMS demonstrated a balanced Th1/Th2 response, enhanced antigen retention in the draining lymph node, and increased expression of cytokines, chemokines and type I interferon-stimulated genes at the injection site and in draining lymph node.
Have the authors measured Th1/Th2 immune profiles?
The results presented in the manuscript text and in Figure 5 reflect the Th1 response, specifically showing the frequency of CD4+ cells producing IFN-g, IL-2 or TNF-a, which are cytokines associated with Th1 activity. The Th2 response was not measured in this study, and this limitation has been discussed on lines 602-607.
How does HI, MNT and T cell immunity correlate within the same participant? Is there also a correlation with side effects?
There are very weak and statistically insignificant positive correlations between HI, MNT and T cell responses. While this holds true across all cohorts, the strongest correlation was found for 0.5mg CMS. The same is true for adverse events: a very weak and statistically insignificant but positive correlation between HI and AEs was observed. Due to the weakness and inconclusiveness of these effects, we decided not to include these data and instead focused on addressing the primary objectives defined in the clinical trial. Ongoing research will explore these correlations further.
Minor comments
Line 15. Define CMS
This has been defined.
Introduction
Please describe CMS in the introduction.
CMS is described in the introduction on line 86-92.
The adjuvant, belonging to a new class of synthetic carbohydrate fatty acid monosulphate-esters (CMS), is immobilized on the surface of the oil droplets within a nano-emulsion of squalane-in-water. Several studies in non-rodent species have demonstrated the potent efficacy of single-shot vaccines containing the CMS-adjuvant, showing comparable or superior antibody titres compared to two immunizations with MF59 adjuvanted vaccines. In ferrets, CMS strongly enhanced humoral responses to a low dose of a licensed vaccine following both initial and booster immunizations.
Methods
How does the final CMS+ Squalene in water emulsion compare to MF-59 formulation? How much of the immunogenicity is due to the emulsion only?
The squalane droplets serve as the vehicle for the CMS molecules. Known adjuvants such as MF59, AS03, SE and GLA-SE contain similar emulsions of squalane or squalene-in-water, but without CMS. Polysorbate 80 stabilizes the squalane-in-water submicron emulsion and acts a surface-active agent, which is also used in several other vaccine adjuvant formulations, including MF59, AS03, SE and GLA-SE. Previous preclinical studies have demonstrated a synergistic effect between squalane and CMS precursors (Hilgers, 2006). Therefore, it is likely that both CMS and squalane contribute to immunogenicity. However, the individual contributions of each component were not determined in this study, and the investigation into the exact mechanisms of action of the adjuvant is currently ongoing research.
Round 2
Reviewer 1 Report
Comments and Suggestions for Authors
The suggestion I made has been revised.
Author Response
The suggestion I made has been revised.
We thank the reviewer again for their time and valuable feedback that led to the improvement of this manuscript. We are happy that the revision was satisfactory.
Reviewer 3 Report
Comments and Suggestions for Authors..
Author Response
..
We thank the reviewer again for their time and valuable feedback that led to the improvement of this manuscript. We are happy that the revision was satisfactory.